

# Habitat association in the critically endangered Mangshan pit viper (*Protobothrops mangshanensis*), a species endemic to China

Bing Zhang[1], Bingxian Wu[1], Daode Yang[1], Xiaqiu Tao[1], Mu Zhang[1], Shousheng Hu[2], Jun Chen[2] and Ming Zheng[2]

[1] Institute of Wildlife Conservation, Central South University of Forestry and Technology, Changsha, Hunan, China
[2] Administration Bureau of Hunan Mangshan National Nature Reserve, Chenzhou, Hunan, China

Corresponding author
Daode Yang, csfuyydd@126.com

## ABSTRACT

Habitat directly affects the population size and geographical distribution of wildlife species, including the Mangshan pit viper (*Protobothrops mangshanensis*), a critically endangered snake species endemic to China. We searched for Mangshan pit viper using randomly arranged transects in their area of distribution and assessed their habitat association using plots, with the goals of gaining a better understanding of the habitat features associated with *P. mangshanensis* detection and determining if the association with these features varies across season. We conducted transect surveys, found 48 individual snakes, and measured 11 habitat variables seasonally in used and random plots in Hunan Mangshan National Nature Reserve over a period of 5 years (2012–2016). The important habitat variables for predicting Mangshan pit viper detection were fallen log density, shrub density, leaf litter cover, herb cover and distance to water. In spring, summer and autumn, Mangshan pit viper detection was always positively associated with fallen log density. In summer, Mangshan pit viper detection was related to such habitats with high canopy cover, high shrub density and high herb cover. In autumn, snakes generally occurred in habitats near water in areas with high fallen log density and tall shrubs height. Our study is the first to demonstrate the relationship between Mangshan pit viper detection and specific habitat components. Mangshan pit viper detection was associated with habitat features such as with a relatively high fallen log density and shrub density, moderately high leaf litter cover, sites near stream, and with lower herb cover. The pattern of the relationship between snakes and habitats was not consistent across the seasons. Identifying the habitat features associated with Mangshan pit viper detection can better inform the forestry department on managing natural reserves to meet the habitat requirements for this critically endangered snake species.

## INTRODUCTION

Many wild animals require multiple habitats to obtain various resources (*Raynor et al., 2017*; *Leite et al., 2018*), which would provide them opportunities for predation, reproduction, and shelter (*Doligez, Danchin & Clobert, 2002*; *Hyslop, Cooper & Meyers, 2009*; *O'Hanlon, Herberstein & Holwell, 2015*). Effective conservation and management of species depends on an understanding of habitat requirements, particularly if these aspects change seasonally. This is especially the case for species susceptible to habitat loss or fragmentation (*Willems & Hill, 2009*; *Ali et al., 2017*; *Mandlate, Cuamba & Rodrigues, 2019*). However, information related to habitat requirements is often scarce when a species has low population densities, narrow and remote habitat, receives little low public attention, and when venomous animals can endanger investigators (*Rechetelo et al., 2016*; *Sutton et al., 2017*; *Leite et al., 2018*). Through investigations into the habitat features associated with a species detection, the important variables that influence habitat association patterns can be found. If the biological resources are limited and patchily distributed across the landscape, the identification and protection of essential habitat components would be critical to population persistence, recovery efforts, and the design of protected areas (*Ali et al., 2017*; *Leite et al., 2018*).

The relationship between the wild animals and their habitat may vary seasonally (*Lunghi, Manenti & Ficetola, 2015*; *Ortega, Mencia & Perezmellado, 2016*). As ectothermic animals, snakes are very sensitive to thermal changes in their external environment, and therefore, the habitat relationship may vary in different seasons based on thermoregulatory requirements (*Richardson, Weatherhead & Brawn, 2006*; *Sprague & Bateman, 2018*). In addition, breeding, prey availability, refugia and other factors are also important factors affecting the seasonal habitat association of snakes (*Harvey & Weatherhead, 2006*; *Sperry & Weatherhead, 2009*; *Gardiner et al., 2015*). Snakes may also choose a preferred habitat factor that is not affected by the seasons, which brings them survival benefits and maximizing resource availability (*Hecnar & Hecnar, 2011*; *Sutton et al., 2017*). For example, snakes may give priority to habitats that are easy to hunt for food and allowing a good place to escape (*Wasko & Sasa, 2012*; *Gardiner et al., 2015*).

The Mangshan pit viper (*Protobothrops mangshanensis*) is the largest species of Viperidae in China (up to 2 m long and 2–4 kg in weight) (*Gong et al., 2013*), but its habitat is limited to just 10,500 ha on a single mountain range. The population of the Mangshan pit viper has been estimated to be less than 500 individuals (*Chen et al., 2013*; *Gong et al., 2013*), and as such, it is classified as an endangered species on the IUCN Red List of Threatened Species, listed in Appendix II of the CITES (Convention on International Trade in Endangered Species of Wild Fauna and Flora) in 2013, and listed as critically endangered on the Red List of China's Vertebrates in 2016 (*Jiang et al., 2016*). Unfortunately, to some extent, the construction of roads and small hydro-power plants, along with the development of tourism, caused destruction, fragmentation, and degradation of the habitat of this species (*Gong et al., 2013*). In addition, illegal harvest of bamboo shoots still occurs in the range of this species, negatively affecting the integrity of habitat composition. All of these may threaten the persistence of this population, as habitat

loss or degradation is a leading driver of wildlife population decline (*Stuart et al., 2004*; *Leite et al., 2018*).

Most studies on the Mangshan pit viper have focused on venom (*Mebs et al., 2006*; *Murakami et al., 2008*; *Valenta, Stach & Otahal, 2012*), identification of individuals (*Yang et al., 2013*), and on population status and distribution (*Gong et al., 2013*). However, little is known about their habitat requirements, which would provide basic information about how the snake meets its needs for survival; therefore, this information is especially crucial in efforts to preserve this at-risk species (*Zhou, 2012*). Since 2012, under the direction of the State Forestry Administration of China, we carried out long-term population monitoring study of the Mangshan pit viper. Exploratory investigations revealed tendencies for this species to occur within primary forest. While associations between this species and habitat factors (e.g., vegetation, fallen log, stream) and the seasonal variation of species-habitat association have been observed, the details had not been rigorously investigated. Therefore, the primary objectives of this study were: (1) identify the habitat features associated with *P. mangshanensis* detection across the study area (2) determine if the association with these features varies across season.

## MATERIALS AND METHODS

### Study area

Hunan Mangshan National Nature Reserve (hereafter referred to as the Mangshan Reserve) is located in Yizhang County, Chenzhou City, Hunan Province, at the northern foot of the Nanling Mountains in China (24°53′00″–25°03′12″N, 112°43′19″–113°00′10″E). Elevations range from 436–1,902.3 m, and the total area covers 198.33 km². Mangshan Reserve lies within the subtropical humid monsoon climatic zone of China, with an average annual temperature, relative humidity, and precipitation of 17.2 °C, of 82.8% and 1,950 mm, respectively. This area features a frost-free period averaging 290 days. The seasons of the Mangshan Reserve are the following: spring = March–April; summer = May–August; autumn = September–October; winter = November–February (*Sun et al., 2011*).
The vegetation type is mainly subtropical evergreen broad-leaved forest in areas <1,000 m a. s.l., with mainly coniferous and broad-leaved mixed forest at elevations >1,000 m a.s.l. (*Fu et al., 2012*). The dominant trees are: *Fagus longipetiolata*, *Michelia foveolate*, *Schima remotiserrata*, *Lithocarpus chrysocomus*, and *Pinus kwangtungensis*. The dominant shrubs are: *Rhododendron fortunei*, *Rhododendron simiarum*, *Vaccinium bracteatum*, *Enkianthus serrulatus* and *Eurya saxicola* f. *puberula*.

### Survey methods

We looked for *P. mangshanensis* individuals by using transect surveys from 2012 to 2016 (*Mazerolle et al., 2007*). We randomly arranged 261 transects in the distribution area of *P. mangshanensis* and the average transect length was 277 m (Table S1). If a randomly selected transect location occurred on an area not accessible to snakes (e.g., open water, traffic corridors, escarpment), a new transect was selected. The total length of all transects was 72 km. Each transect was investigated twice in the daytime (12:00–15:00) and once at night (20:00–23:00), and each transect was repeatedly investigated in three different

months: April, July and October. The observers were divided into three groups, with five observers in each group. In each transect, all five observers traveled in a single transect along a line at a speed of about 0.3 km/h, one by one, with 5 m spacing between each pair of observers. The observers recorded all detected individuals on both sides to 5 m width in a 10 m width transect. After a snake was discovered, we recorded the GPS location accurately using a global positioning system (GPS) unit (Beijing UniStrong Science and Technology Co., Ltd., Beijing, China). We used head patch pattern as a reliable biometric character to recognize Mangshan pit viper individuals (*Yang et al., 2013*) and recorded each individual's ID. Based on field surveys conducted from 2012 to 2016, we found 48 individual snakes and identified 83 locations for seasonal habitat studies (20 sites surveyed in spring, 31 in summer and 32 in autumn; Fig. 1) using 10 m × 10 m plots (fourth-order). Plots used by snake individuals (used plots) were placed with the location used by *P. mangshanensis* as the center point. To compare used and random habitat, we conducted habitat studies at used plots and random plots (*Keating & Cherry, 2004*; *Johnson et al., 2006*). The direction and distance (between 50 and 150 m) of the random plot from each used plot were determined using a random number generator (*Sprague & Bateman, 2018*). If the random plot occurred in an area that was not accessible to snakes (e.g., open water, traffic corridors, escarpment), a new location was determined. Habitat variables were measured in used and random plots in April (spring), July (summer) and October (autumn) of 2015 and 2016. Some snake observations predated collection of the habitat data. In order to maintain the consistency of variables, we collected variable data in the same month within three years. However, such variables had likely changed since these individuals were detected. Therefore, this study can only analyze the association between the *P. mangshanensis* detection and their habitats to a certain extent. The study was performed in accordance with the recommendations of the Institution of Animal Care and the Ethics Committee of Central South University of Forestry and Technology (approval number: CSUFT-871965). Permission for fieldwork was obtained from the Administration Bureau of Hunan Mangshan National Nature Reserve (permit number: MSNR-12317).

## Habitat variables

Based on a review of the current literature and data from our previous research (Fig. 2), we identified 11 important habitat variables for *P. mangshanensis* (*Baxley, Lipps & Qualls, 2011*; *Gardiner et al., 2015*; *Buchanan et al., 2017*; *Sutton et al., 2017*; *Sprague & Bateman, 2018*). Habitat variables were measured as follows. We estimated canopy cover using a sighting tube with crosshairs at one end (*Winkworth & Goodall, 1962*; *Sperry & Taylor, 2008)*, and recorded the number of canopy hits out of 20 random sightings within each 1-m$^2$ quadrat at the four corners and the center of the plot. These values were averaged, and then the average value was multiplied by 5 to estimate percent canopy cover. Herb cover, herb height, leaf litter cover, shrub density, and shrub height were also measured within five 1-m$^2$ quadrats at the plot, with an average calculated for each variable. We measured the herb cover within each 1-m$^2$ quadrat. If the shape of the herb cover area was irregular, we roughly divided it into several regular shapes and counted the

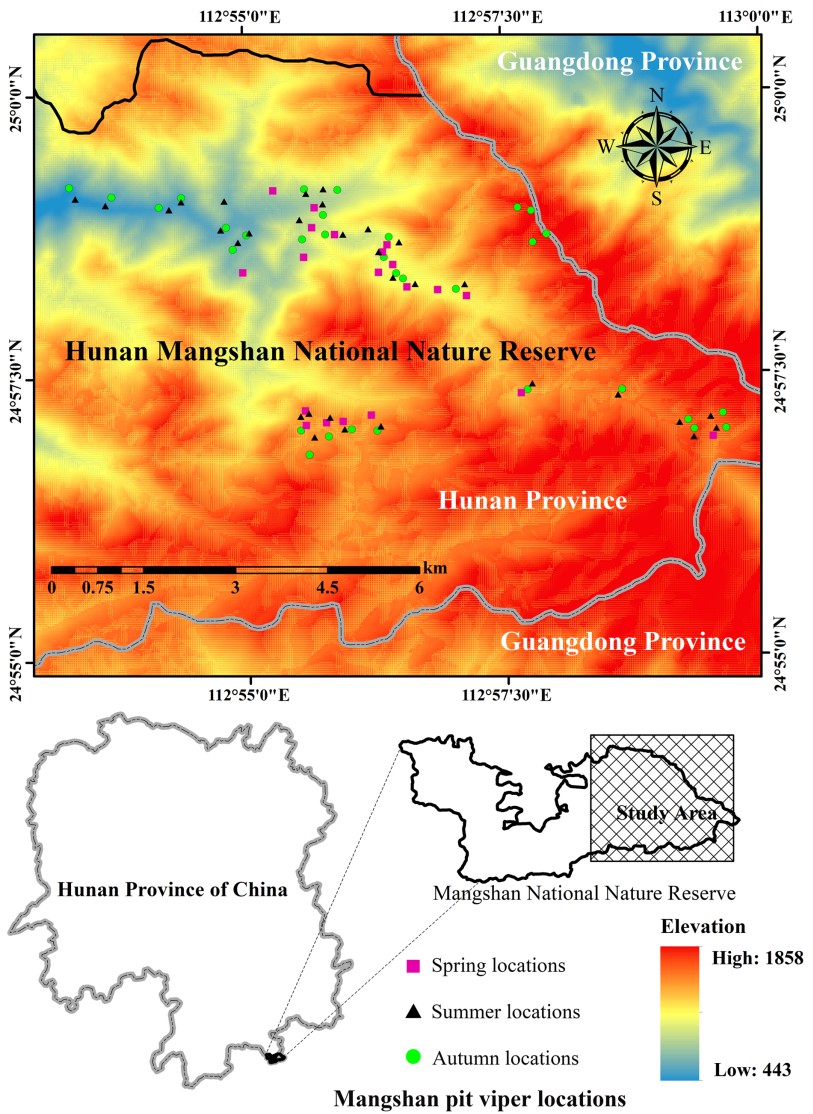

**Figure 1 The study area and the location data of *Protobothrops mangshanensis*.** All of the snake individuals were found only in the eastern part of the Hunan Mangshan National Nature Reserve. Figure source credit: Xiaofeng He and Simin Wu.

sum of the areas of several regular shapes. Then, we calculated the percentage of the herb cover area within each 1-m² quadrat to represent the herb cover. Herb height was the average of maximum height of herbs each 1-m² quadrat. Leaf litter cover was measured by a method similar for herb cover. Shrub density was measured as the total number of shrubs stems within each 1-m² quadrat. Shrub height was the average height of shrubs within each 1-m² quadrat. Distance to water was measured as the linear distance between the center of plot and the nearest permanent stream (channel width > 1 m) using a Nikon Forestry 550 laser rangefinder. Elevation was obtained at the center of the plots by Orux Map software. Fallen log density was calculated as the number of fallen logs in each plot (diameter > 4 cm, length > 0.5 m). Slope gradient was measured from the lowest to the highest point in each plot using a Nikon Forestry 550 laser rangefinder (Nikon, Tokyo,

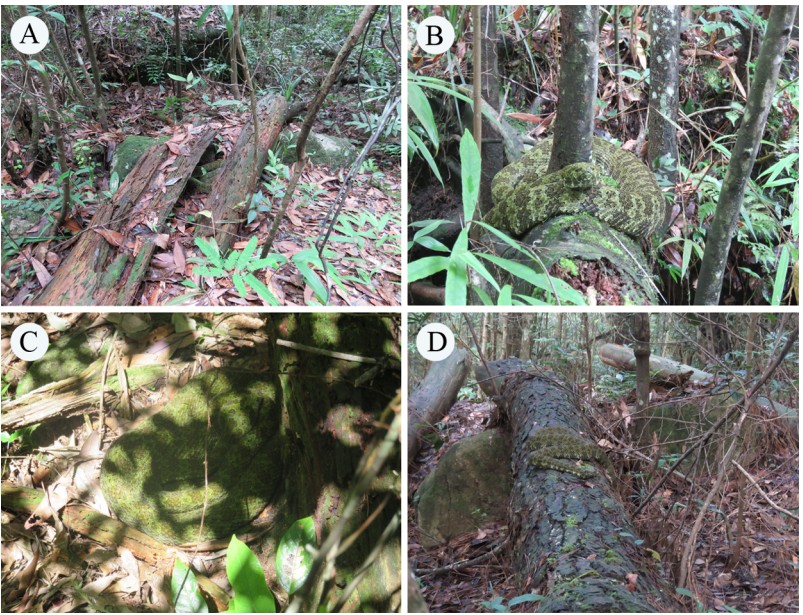

**Figure 2 The Mangshan pit viper (*Protobothrops mangshanensis*) and its habitats.** The body color of the Mangshan pit viper blends well into the surrounding environment. (A) Typical habitats of the Mangshan pit viper; individual Mangshan pit vipers (B) on fallen log, (C) selects a basking spot in sunshine and enhances its body temperature, (D) crawls on a fallen log.

Japan). Tree density was calculated as the number of trees stems in the plot (diameter at breast height > 4 cm).

## Statistical analyses

We used the Random Forests method in R v 3.5.1 to examine the relationship between use–random plots and each habitat variable (*Breiman, 2001*; *Liaw & Wiener, 2002*; *R Development Core Team, 2018*). We included all the variables in our analysis because none of the 11 microhabitat variables chosen were highly correlated ($r < 0.7$) (*Gardiner et al., 2015*). The random forest method was based on bootstrap samples of the training data set and combined many different trees. In a typical bootstrap sample, about 63% of the original observations occurred at least once. We called the observations in the original data set that do not occur in a bootstrap sample as out-of-bag observations and considered random selection of variables when choosing splits in each node. The random forest method was rarely over-fitted and can provide efficient predictions with large numbers of independent variables (*Breiman, 2001*). In this study, we used partial dependance plots to graphically characterize relationships between each predictor variable and predicted probabilities of *P. mangshanensis* detection obtained from the Random Forest analysis (*Hastie, Tibshirani & Friedman, 2005*).

In order to further identify the habitat variables associated with Mangshan pit viper detection, we used generalized linear mixed models (GLMMs) via the lme4 package (*Bates et al., 2015*) in R v.3.5.1 (*R Development Core Team, 2018*). We created a dataset where (1) represented "used" plots and (0) represented "random" plots. We used a mixed-models

approach to account for the non-independence of habitat samples for individual snakes (random effect). The variables used to build the models were the top seven variables with high ranking from the Random Forests analysis. A model for each combination of variables was created with each model including the mixed-effect function. We used the glmer function in the lme4 package to build the model and the model.avg function in the glmulti package to compare with the previous models in R v.3.5.1 (*R Development Core Team, 2018*). We used the predict function to predict the results. All possible models were considered (R package "rJava, glmulti, and MuMIn"). The models were screened by Akaike's Information Criterion (AIC) (*Burnham & Anderson, 2002*). The "best" model had a ΔAICc = 0, but we also considered all models with a ΔAICc < 2 (*Burnham & Anderson, 2002*; *Mazerolle, 2006*).

To determine if the relationship between Mangshan pit viper detection and the habitat features varied across season, we created a dataset where (0) represented all "random" plots, (1) represented "spring" plots, (2) represented "summer" plots, (3) represented "autumn" plots. We first used the aov function for ANOVA, and then used the LSD.test function in the agricolae package to perform bonferroni correction to obtain the final $p$ value. The formula for bonferroni correction is $p \times (1/n)$, where $p$ is the original threshold and $n$ is the total number of inspections. These data met assumptions of normality and equal variance. Tests were considered significant at $p < 0.05$. All statistical analyses were conducted in R v 3.5.1 (*R Development Core Team, 2018*).

## RESULTS

We used the Random Forest model with out-of-bag samples to evaluate the importance of predictor variables (Fig. 3). By measuring the variable importance, we computed the total decrease in node impurities (Gini index) for each variable given by the splitting of the variable. Highly ranked variables were fallen log density, shrub density, leaf litter cover, herb cover, distance to water, shrub height, and herb height (Fig. 3). We checked the response curves between predicted values and highly ranked important variables using a partial dependance plot (Fig. 4). For fallen log density (Fig. 4A) and shrub density (Fig. 4B), when the value of these variables were larger, relatively high predicted values were shown. When fallen log density > 13 and shrub density > 15, the predicted value tends to be stable. When the variables of distance to water < 23 m (Fig. 4C) and herb cover < 15% (Fig. 4D), relatively high predicted values were acquired. Leaf litter cover had an optimal range of 70–80% (Fig. 4E). Moreover, when herb height < 21 cm (Fig. S1A), and shrub height > 200 cm (Fig. S1B), relatively high predicted values were observed.

Through the GLMMs, we constructed four optimal models and the "best" model with min ΔAICc showed that leaf litter cover, distance to water, fallen log density, herb cover and shrub density associated with *P. mangshanensis* detection (Table 1), which was also similar for the Random Forest model analysis. Therefore, the snake detection associated with such habitat features, which had a relatively high fallen log density and shrub density, moderately high leaf litter cover, near water (permanent stream) and lower herb cover.

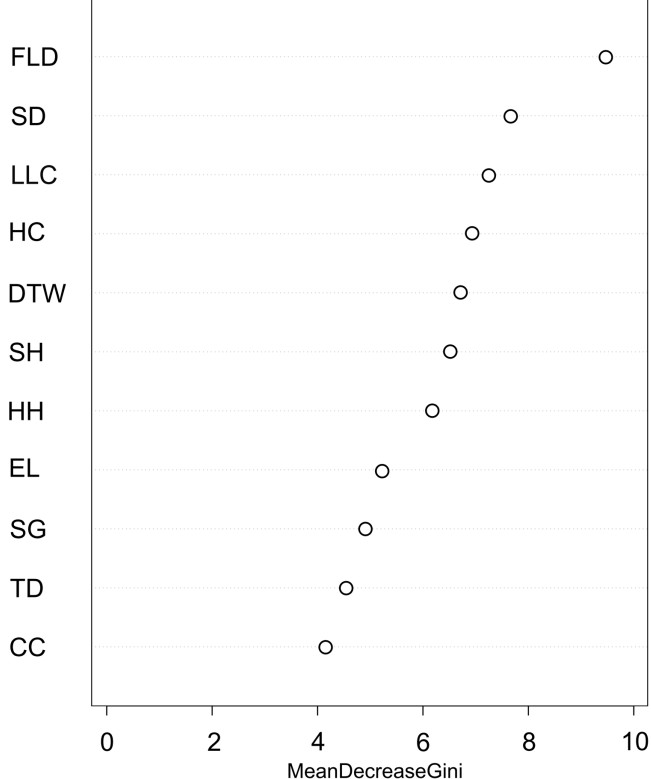

**Figure 3 Variable importance plot for predictor variables from Random Forest classifications used for predicting the occurrence of Mangshan pit viper.** FLD, fallen log density; SD, shrub density; LLC, leaf litter cover; HC, herb cover; DTW, distance to water; SH, shrub height; HH, herb height; EL, elevation; SG, slope gradient; TD, tree density; CC, canopy cover.

Season influenced the relationship between Mangshan pit viper detection and the habitat features (Table 2). During the spring, Mangshan pit viper detection was positively related to fallen log density. In summer, Mangshan pit viper detection was related to such habitats with high fallen log density, high canopy cover, high shrub density and high herb cover. Unlike spring and summer, in autumn snakes generally occurred in habitats near water with high fallen log density and shrubs height.

## DISCUSSION

Our approach of using transect surveys to discover Mangshan pit viper provides an assessment of the habitat features associated with the detection of this critically endangered species across the study area. In this study, Mangshan pit viper detection was related to fallen log density, shrub density, leaf litter cover, herb cover and distance to water. These habitat features may provide them with necessary survival resources, such as refugia and water. In addition, the relationship between Mangshan pit viper detection and the habitat features was also influenced by the seasons.

Snakes usually choose rocks, vegetation, and burrows as refugia (*Webb, Shine & Pringle, 2005*; *Hyslop, Cooper & Meyers, 2009*; *Bruton et al., 2014*). The need for thermoregulation and the location of potential prey influenced the site selection of snakes seeking refugia

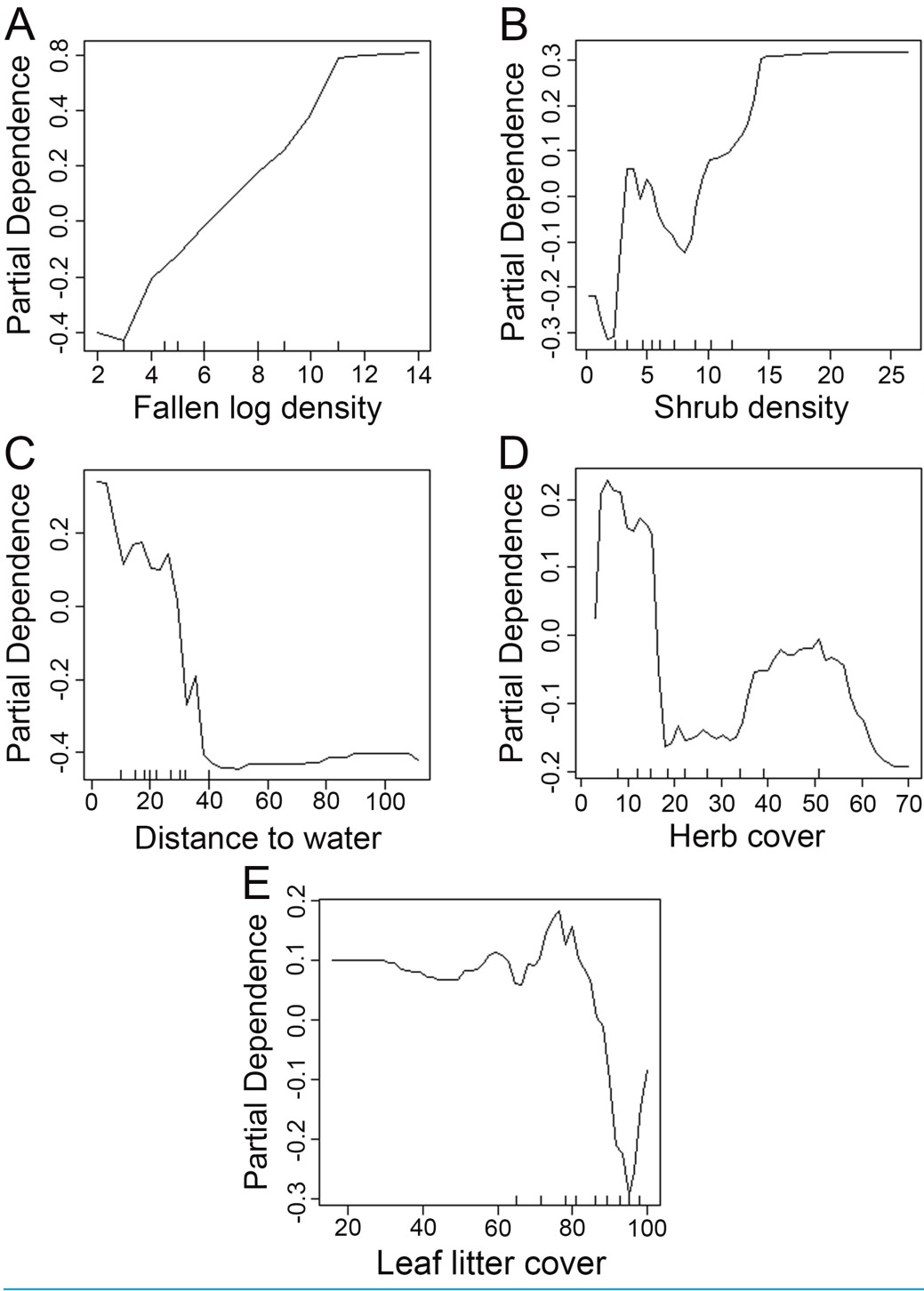

**Figure 4 Partial dependance plots for high ranked predictor variables for Random Forest predictions of the occurrence of Mangshan pit viper.** Partial dependance is the dependance of the probability of occurrence on one predictor variable after averaging out the effects of the other predictor variables in the model. (A–E): the dependance of the probability of occurrence on fallen log density, shrub density, distance to water, herb cover and leaf litter cover, respectively.

**Table 1 Models for predicting habitat relationships of Mangshan pit viper (*Protobothrops mangshanensis*).**

| Model ID | Models | k | ΔAICc | Weight |
|---|---|---|---|---|
| 1 | DTW + FLD + HC + LLC+ SD | 7 | 0.00 | 0.29 |
| 2 | DTW + FLD + HC + LLC + SD + SH | 8 | 0.36 | 0.25 |
| 3 | DTW + FLD + HC + HH + LLC + SD +SH | 9 | 0.89 | 0.19 |
| 4 | DTW + FLD + HC + HH + LLC + SD | 8 | 1.44 | 0.14 |

Note:
DTW, distance to water; FLD, fallen log density; HC, herb cover; HH, herb height; LLC, leaf litter cover; SD, shrub density; SH, shrub height. Models were ranked according to Akaike's Information Criterion (AIC).

**Table 2 Descriptive statistics and comparison of values of eleven ecological variables in used versus random plots for habitat of the Mangshan pit viper (*Protobothrops mangshanensis*) in three different seasons (mean ± SE).**

| Ecological variable | Random plots (n = 83) | Spring (n = 20) Used plots | Rel. | p-Val. | Summer (n = 31) Used plots | Rel. | p-Val. | Autumn (n = 32) Used plots | Rel. | p-Val. |
|---|---|---|---|---|---|---|---|---|---|---|
| Canopy cover (%) | 77.5 ± 1.2 | 67.0 ± 3.9 | – | 0.191 | 84.8 ± 1.1 | + | 0.037 | 73.9 ± 2.1 | – | 0.501 |
| Distance to water (m) | 29.9 ± 2.2 | 16.7 ± 1.8 | – | 0.858 | 26.6 ± 5.2 | – | 0.127 | 24.4 ± 1.0 | – | 0.026 |
| Elevation (m) | 1025 ± 30 | 1039 ± 38 | + | 0.999 | 992 ± 49 | – | 0.825 | 1064 ± 57 | + | 0.974 |
| Fallen log density (number/100 m$^2$) | 5.0 ± 0.3 | 7.9 ± 0.8 | + | 0.001 | 7.4 ± 0.6 | + | <0.001 | 7.2 ± 0.4 | + | 0.007 |
| Herb cover (%) | 28.4 ± 1.8 | 17.3 ± 2.1 | – | 0.711 | 30.1 ± 3.2 | + | 0.001 | 21.7 ± 2.5 | – | 0.863 |
| Herb height (cm) | 31.8 ± 2.0 | 19.6 ± 1.2 | – | 0.067 | 30.3 ± 3.7 | – | 0.997 | 30.0 ± 3.2 | – | 0.741 |
| Leaf litter cover (%) | 84.2 ± 1.4 | 74.4 ± 4.8 | – | 0.988 | 84.2 ± 2.2 | + | 0.905 | 83.9 ± 1.8 | – | 0.450 |
| Shrub density (trees/m$^2$) | 6.2 ± 0.4 | 6.8 ± 0.6 | + | 0.986 | 7.9 ± 1.1 | + | 0.001 | 7.8 ± 0.6 | + | 0.499 |
| Shrub height (cm) | 162.1 ± 4.2 | 136.5 ± 5.6 | – | 0.814 | 187.9 ± 6.1 | + | 0.992 | 177.2 ± 9.4 | + | 0.025 |
| Slope gradient (°) | 27.8 ± 1.6 | 17.7 ± 1.7 | – | 0.956 | 29.6 ± 2.7 | + | 0.889 | 28.2 ± 2.4 | + | 0.862 |
| Tree density (trees/100 m$^2$) | 18.1 ± 0.8 | 18.5 ± 1.4 | + | 0.826 | 18.5 ± 1.4 | + | 0.957 | 19.5 ± 1.5 | + | 0.613 |

Note:
Direction of habitat association shown as positive or negative relative to random plots.

(*Whitaker & Shine, 2003*; *Webb, Shine & Pringle, 2005*). A lack of adequate refugia can perturb behaviors, increase stress levels, and thus alter physiological performance (e.g. digestive, immune, or reproductive functions) for snakes (*Bonnet, Fizesan & Michel, 2013*). Resources may be unevenly distributed in space in habitats within the home range of animals, so that the animals must move about to seek the best locations, which can influence their acquisition of nutrients (*Sperry & Weatherhead, 2009*), perfect mimicry (*O'Hanlon, Herberstein & Holwell, 2015*; *Skelhorn & Ruxton, 2013*), and help with thermoregulation (*Ortega & Pérez-Mellado, 2016*).

Our data indicated that the habitat element of fallen log density was associated with *P. mangshanensis* detection, and this relationship was consistent in all three seasons analyzed here. The Mangshan pit viper is an ambush feeder; they lie in wait until prey appear. Then, they may use caudal luring (a white tail that resembles vermiform) to attract the prey at that point. Furthermore, the body color and markings of the Mangshan pit viper and the lichen on fallen logs is similar. Their camouflage allows them to blend into the habitats traversed by their prey during the preys' foraging movements, thereby

potentially increasing the likelihood of an encounter. We predict that the snakes appear near fallen logs that maximize the efficacy of their deceptive signal and the likelihood that signal receivers are successfully deceived, which is an optimal foraging strategy and under optimal foraging theory (*O'Hanlon, Herberstein & Holwell, 2015*). Therefore, fallen logs or other coarse woody debris may be important components of snake habitat (*Vanek & Wasko, 2017*). In recent years, the Administration Bureau of Mangshan Reserve has carried out tourism in the experimental zone of the Mangshan Reserve. However, there was a certain overlap between the distribution area of *P. mangshanensis* and tourism development area. Furthermore, the Administration Bureau was not aware of the association between the fallen logs and *P. mangshanensis*. In order to facilitate the patrol of the preserve managers and tourists' sightseeing, the Administration Bureau cleared some fallen logs, which may damage the refugia of snakes.

The shrub density is also a highly ranked variable. The partial dependance plot showed that *P. mangshanensis* were more likely to occur in habitats with relatively high shrub density (Fig. 4). Shrub habitats may be often visited by small mammals (*Gardiner et al., 2015*) that may provide a prey source for snakes. In addition, higher shrub density may affect detection of *P. mangshanensis* by predators and provide snakes with the convenience of thermoregulation. The areas of higher leaf litter cover were also associated with the Mangshan pit viper detection. Previous studies have also revealed that snakes avoided bare ground (*Sperry et al., 2009*; *Baxley, Lipps & Qualls, 2011*; *Gardiner et al., 2015*). Leaf litter cover can keep the ground temperature relatively stable and provide conditions for thermoregulation (*Buchanan et al., 2017*). In areas of bare soil, the temperature changes greatly, which may exceed the tolerance limit of snakes. Compared with the highly ranked variables above, lower herb cover was positively correlated with the probability of snake detection. Many snakes choose dense vegetation as refugia, such as shrubs and herbs (*Baxley, Lipps & Qualls, 2011*; *Shew, Greene & Durbian, 2012*). Mangshan pit viper may prefer shrubs to herbs for refugia, or snakes may be more detectable when herb cover is relatively low. Water availability and distribution are important determinants of behavior and habitat selection in snakes (*Halstead, Wylie & Casazza, 2010*). Our data indicated that the probability of snake detection was positively correlated with a relatively short distance to water, which was consistent with other studies in that the proximity to water is important for snakes (*Brito, 2003*; *Halstead, Wylie & Casazza, 2010*; *Sprague & Bateman, 2018*). In addition, by employing camera traps we observed that small mammals were more abundant in habitats that were relatively close to water (B. Zhang, X. Ding & D. Yang, 2018–2019, personal observations). Such habitats might provide improved foraging opportunities to Mangshan pit viper. However, the development and maintenance of roads, paths, and scenic spots for the service of tourism in Mangshan Reserve may affect the flow of some streams and even change their spatial distribution, which may indirectly affect the habitat of *P. mangshanensis*.

The relationships between species and habitats may not be consistent across the seasons (*Brito, 2003*; *Hyslop, Cooper & Meyers, 2009*; *Sprague & Bateman, 2018*). The change of habitat association patterns of species can be explained by two hypotheses: species may choose different habitats in different seasons (selection change hypothesis);

the characteristics of the external environment needed for the survival of a species may change in different seasons (environmental change hypothesis) (*Lunghi, Manenti & Ficetola, 2015*). For the selection change hypothesis, the behavioral activities (e.g., reproduction, foraging, hibernation) in different time periods and/or life stages explain the changing association between the species detection and habitat features (*Brambilla & Saporetti, 2014*). According to the environmental change hypothesis, temporal variation that exists for the habitat can affect the association between species detection and the habitat features (*Kearney et al., 2013*). Our data showed that the association between *P. mangshanensis* detection and habitat features was seasonal. The variant association may be determined by seasonal variation in the environment. For example, the ambient temperature was higher in summer so that snakes may be forced to occupy a cool habitat with dense vegetation in order to follow the changing environmental conditions. However, it is also possible that this variant was determined by changes in the preferred habitat. For example, *P. mangshanensis* may prefer habitats near water, with high fallen log density and shrubs height, in autumn. In order to truly grasp the mechanism of seasonal shift of the association between *P. mangshanensis* detection and the habitat features, we need to use radio tracking technology for further research (*Sprague & Bateman, 2018*).

For this study, the change of detection probability may have an important impact on Mangshan pit viper discovery. As a cryptic reptile with camouflage color patterns, *P. mangshanensis* individuals are difficult to detect in their natural environments. In addition, detectability can depend on many factors, such as the sampling method selected, sampling effort, habitat type, and the experience of the observers (*Mazerolle et al., 2007*). In order to deal with the change of detection probability, we randomly placed all line transects and repeated the survey three times in different months for each transect. Before the formal investigation, we trained the observers to unify the investigation procedures. However, the ten meters wide transect includes many opportunities for snakes to remain in hiding and undetected. Snakes would likely be less detectable under dense cover than in open habitat. Further, field observations confirmed strong associations of snake individuals with fallen logs. So, our observers may be more inclined to search more carefully in and around these logs than they search in other habitats. What is more, the sex, age, reproductive status or even metabolic condition (i.e., hungry or digesting) of a snake may affect its habitat selection (*Du, Webb & Shine, 2009*; *Sutton et al., 2017*; *Sprague & Bateman, 2018*). For example, gravid females may be more likely to be exposed due to thermoregulatory needs (*Sprague & Bateman, 2018*). Therefore, the imperfect detection method used in our surveys may lead to a more in-depth analysis of the habitat of exposed snakes. However, these factors affecting detection are uncontrollable.

The Administration Bureau of Mangshan Reserve uses strict management techniques in the reserve. From 2012 to 2014, to limit disturbance to snake behavior, we were only allowed to conduct transect surveys. In 2015 and 2016, we were approved to collect habitat data only about 1–2 days after snakes had moved from a location. In addition, in 2015, we collected habitat variable data of individual snakes found between 2012 and 2014 (in the same month), which lagged behind snake detection for one to three years. While some habitat variables remain consistent over time (e.g., distance to water, elevation,

fallen log density, slope gradient); others likely vary (e.g., aspects associated with vegetation including canopy cover, herb cover and height). Mangshan Reserve was designated as a forest reserve in 1958 and is a typical representative of subtropical broad-leaved forest in China, which has a large area of primary forest with high plant diversity (*Huang et al., 2012*). The plant community of the primary forest in the Mangshan Reserve has succeeded as a stable climax community over time (*Li et al., 2020*). Furthermore, all Mangshan pit vipers were all detected in primary forest. Therefore, we speculate that the change of vegetation in the study area would be negligible over a three years period.

In transect surveys, we observed that the Mangshan pit viper generally occurred in broad-leaved forest. Therefore, we only studied the relationship between snakes and microhabitats in broad-leaved forest. To the best of our knowledge, our study is the first to report the Mangshan pit viper's use of broad-leaved forest habitat and the first to investigate details of microhabitat association by this snake species. The study of fine-scale habitat features selected by snakes is important for mastering our understanding of the available habitat structure (*Hecnar & Hecnar, 2011*; *Gardiner et al., 2015*). However, snakes may exhibit varied habitat selection patterns at different spatial scales (*Sutton et al., 2017*). Assessing habitat selection at one spatial scale may result in weak inferences regarding species-habitat relationships. Therefore, we urge multi-scale habitat evaluations of Mangshan pit viper should be conducted as soon as possible to provide more information on management recommendations for protected snake populations.

## CONCLUSIONS

The habitat features associated with Mangshan pit viper detection were relatively high fallen log density and shrub density, moderately high leaf litter cover, proximity to water (permanent stream), and relatively low herb cover. The association between snake detection and the habitat features was seasonal. Based on the habitat requirements of *P. mangshanensis* and the current management status of the Mangshan Reserve, we offer the following suggestions for the continued conservation of this critically endangered snake species. (1) Some of the natural refugia used by this species have been destroyed by the construction of hydropower stations, man-made water channels and tourist trails, so methods should be explored to rehabilitate the lost or degraded habitat of *P. mangshanensis* by building artificial refugia that mimic the appropriate physical characteristics of fallen log refugia associated with the detection of snakes. (2) A scientifically sound plan should be designed to prevent tourism from damaging to vegetation and changes in the distribution of streams.

## ACKNOWLEDGEMENTS

We thank Prof. Richard Shine and Dr. Melanie Elphick for providing kind help with references and manuscript preparation, and Dr. Yayong Wu, for providing valuable advice on this manuscript. We thank Xiaofeng He and Simin Wu for making the distribution map of study area. We express special appreciation to Guoxing Deng, Jianguo Tan, Tianbin Liu, Desheng Chen, and Yuanhui Chen from Administration Bureau of Hunan Mangshan

National Nature Reserve for their assistance in field work. We are much indebted to the editor and referees for their valuable comments.

### Funding

This work was supported by the National Natural Science Foundation of China (No. 31472021), the project for Endangered Wildlife Investigation, Supervision and Industry Regulation of the National Forestry and Grassland Bureau of China (No. 2019072-HN-001), and the project for Endangered Wildlife Protection of Hunan Forestry Bureau of China (No. HNYB-2019001). The funders had no role in study design, data collection and analysis, decision to publish, or preparation of the manuscript.

### Grant Disclosures

The following grant information was disclosed by the authors:
National Natural Science Foundation of China: 31472021.
Endangered Wildlife Investigation, Supervision and Industry Regulation of the National Forestry and Grassland Bureau of China: 2019072-HN-001.
Endangered Wildlife Protection of Hunan Forestry Bureau of China: HNYB-2019001.

### Competing Interests

The authors declare that they have no competing interests.

### Author Contributions

- Bing Zhang conceived and designed the experiments, performed the experiments, analyzed the data, prepared figures and/or tables, authored or reviewed drafts of the paper, and approved the final draft.
- Bingxian Wu conceived and designed the experiments, performed the experiments, analyzed the data, prepared figures and/or tables, authored or reviewed drafts of the paper, and approved the final draft.
- Daode Yang conceived and designed the experiments, performed the experiments, prepared figures and/or tables, authored or reviewed drafts of the paper, contact authoritative experts to make constructive suggestions on the revision of this manuscript, and approved the final draft.
- Xiaqiu Tao performed the experiments, analyzed the data, prepared figures and/or tables, and approved the final draft.
- Mu Zhang performed the experiments, analyzed the data, prepared figures and/or tables, and approved the final draft.
- Shousheng Hu performed the experiments, prepared figures and/or tables, and approved the final draft.
- Jun Chen performed the experiments, prepared figures and/or tables, and approved the final draft.
- Ming Zheng performed the experiments, prepared figures and/or tables, and approved the final draft.

## Animal Ethics

The following information was supplied relating to ethical approvals (i.e., approving body and any reference numbers):

This study was performed in accordance with the recommendations of the Institution of Animal Care and the Ethics Committee of Central South University of Forestry and Technology (CSUFT-871965).

## Field Study Permissions

The following information was supplied relating to field study approvals (i.e., approving body and any reference numbers):

Permission for fieldwork was obtained from the Administration Bureau of Hunan Mangshan National Nature Reserve (permit number: MSNR-12317).

## Data Availability

The raw data are available in a Supplemental File.

## Supplemental Information

Supplemental information for this article can be found online at http://dx.doi.org/10.7717/peerj.9439#supplemental-information.

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
