# Peer review of "Habitat association in the critically endangered Mangshan pit viper (Protobothrops mangshanensis), a species endemic to China"

_PeerJ, doi:10.7717/peerj.9439_

## Round 0.1 · original submission · Major Revisions

Three reviewers have now evaluated your manuscript and find your study to be generally well-done. However, there are some comments that need to be addressed from each reviewer.

In particular, I highlight that the reviewers 1) identified the need to account for detection probabilities in your analysis and 2) would like to see more elaborate details on your methodology or would like some additional or alternative analyses.

I also agree with the reviewers that the readability of this manuscript is generally very good but there are a some grammar and sentence structure issues that would make the manuscript easier to read. Please pay particular attention to the legibility of the manuscript prior to your next submission.

I note that the reviewers really enjoyed the photographs of your study animal! Thank you for producing a good study on a great species that is perhaps not well known to many of us. I look forward to the revised version of your manuscript.

·

Basic reporting

The authors present an interesting study about an imperiled and secretive snake. The experimental design has produced novel information about habitat selection of this species, although I note below there are some caveats that may restrict interpretation.

The English is excellent but could be improved to more precisely communicate meaning and ensure correct grammar, for example lines 43-46 and 54-56.

Lines 68-74: The objectives/hypotheses imply there is a cyclical nature to habitat selection coincident with the seasons. However, “constantly changing” implies to me there is no order to habitat selection.

Experimental design

Lines 89-101: More information is needed to explain how transects were designed, where they were placed, and whether they changed. The reader needs to be assured that there is no bias in where the transects were placed that would affect purported habitat selection results.

Line 94: A number of studies have shown reptiles select habitat at multiple scales. What does a plot of 10x10 m represent to the snake and how/why was this size plot chosen by the researchers?

The sex, age, reproductive status or even metabolic condition (i.e., hungry or digesting) of a snake may affect how it selects habitat. The authors should note this was not controlled for.

It is critical to consider how variation in detection probability has affected results (for example, see Mazerolle et al. 2009). It is possible that snakes were easier to observe in certain habitats (perhaps with less vegetation); the current analysis will indicate that snakes selected these habitats even though they could be selecting multiple other habitats where they were more difficult to observe. Further, if the authors already suspected that these vipers are associated with logs (there is reference to an independent tracking study) they may be more inclined to search in and around these logs than they are in other habitats.

I’m concerned that there were nearly as many habitat variables (14) as there were plots (20-32). Can the authors demonstrate that this is a valid statistical analysis? Even after correlated variables were removed, the number remaining seems relatively high.
I am not qualified to evaluate the fine points of the analyses, but some questions were raised as I was reviewing associated parts of the manuscript. Some more information about the analyses and specifically what each were evaluating would be helpful. For example:

Line 148: I’m unsure if three separate analyses were conducted for each season or whether season was a variable included in one analysis. Some more information related to the analytical framework would be helpful. For example, would the authors have analyzed habitat selection by season if the stepwise discriminant function analysis had not indicated that they could be distinguished?

The authors have shown that what is available to the snakes is different from what is selected by the snakes within each season, but it’s unclear whether they analyzed whether what was available differed by season, which seems necessary to tease out whether changes in habitat selection by season are due to decisions made by the snake or seasonal changes in habitat.

Validity of the findings

Line 214-215: Did habitat selection change because of habitat heterogeneity or because they were selecting different habitats based on their physiological needs and/or reproductive status, for a couple examples.

Line 251-252: This is unclear. Did the availability of water change by season? If this is the case, how can the authors know whether they are analyzing changes in habitat selection or changes in habitat availability across different seasons?

Lines 252: What tracking study were the authors referring to? Are the authors implying that snakes selected habitats near water primarily because of foraging opportunities? Do they eat less in the summer?

Line 285: this mention of poachers here in the section about seasonal differences is a non sequitur.

Lines 288-291: This sentence seems redundant.

Lines 275-301: This section seems redundant with information presented elsewhere. I suggest the authors consider condensing and combining the sections of the discussion, first explaining that habitat selection changed by season and then providing examples why it might vary, including different needs related to shelter, thermoregulation, etc.

Lines 313-322: There is little background information or context provided for these conclusions and no clear link to the research and results described in this study.
Table 2: It is unclear what ‘correctly classified’ means.

Mazerolle, M.J. et al. 2009. Making great leaps forward: accounting for detectability in herpetological field studies.

·

Basic reporting

This article could use some proof-reading and copy-editing to improve the sentence structure and word choice. I have included some suggestions in the attached PDF.

Experimental design

The methods seem appropriate, but I feel some elaboration is needed. In general, more information on the transects is needed, clarity on sample sizes, and more detail about the statistical tests. This is detailed in my attached PDF.

Validity of the findings

The main facet I think needs to be included is a discussion of detection probability. As these snakes were encountered opportunistically along transects, do the results represent an overall view of habitat selection, or rather is the selection biased towards snakes that are basking and thus easily (relatively) observed? Also, how does the time of day impact these observations? On hot days, are snakes more or less likely to be out basking, etc.

In my own research (in prep), I found that snakes located via radio-telemetry were more likely to be found in areas of dense vegetation relative to snakes found during timed meanders, and that many snakes were completely concealed and thus would not have been found in transect surveys.

I think the results of this paper are still valid and important, but they need to be framed in the context of imperfect detectability.

Additional comments

This is a great paper that documents important natural history and habitat selection data that will be important for conservation. Please see my attached changes for suggestions to improve the manuscript.

·

Basic reporting

Introduction – The introduction is generally well-organized and flows well. The authors start broadly and then focus down to the species of concern and objectives of the study, and reference appropriate studies throughout. There are several aspects, however, that could be improved.

1. I recommend that the grammar and style of this section be carefully edited, as there are many instances where the terminology used within a sentence is not quite stylistically cohesive. The pieces are mostly there, but the connections and writing style could be improved. For example, lines 40-41 I would change to: Effective conservation and management of species depends on an understanding of habitat use and selection, particularly if these aspects change seasonally. This is especially the case for species susceptible to habitat loss or fragmentation. Asking a colleague familiar with the topics explored in the background to reshape sentences would serve to greatly strengthen this section, in my opinion.

2. The first paragraph could benefit from a direct connection to why an understanding of habitat use is critical to species conservation and management. This really sets the stage for the whole introduction and importance of the study. The pieces are there, but the direct connections are not, so a reader unfamiliar with these fields may be left guessing at the significance. For example, such data could assist in the establishment of appropriately sized or placed reserves. Or…if the resources are limited and patchily distributed across the landscape, the identification and protection of movement corridors would be critical to population persistence. Something like this to really emphasize the connection for the reader. I say this, because all species require multiple resources for their survival, but the real issue relates to actually attaining those resources in a landscape shaped by humans.

3. The authors do a good job establishing the need for this particular study in terms of how it adds to our current understanding of Mangshan Pitviper ecology (lines 63-67). This paragraph could be further strengthened by mentioning current threats to population persistence (does habitat fragmentation or alteration affect movement or survival? What role does illegal collection play? - mentioned later in the discussion). Include anything relevant to the motivation to collect the habitat data that you did in this study. Specifically, any mention to the interest in testing for seasonal effects is currently missing in this section. More context needs to be provided in this regard.

Detailed comments:
• Line 38, habitat or habitat types
• Line 46, change poisonous to venomous
• Line 51, instead of retreating, use the term refugia
• Line 58, I think that mentioning this species is limited to just a single mountain range would be useful for the reader (change to: limited to just 10,500 ha on a single mountain range).
• Table 1 caption states twelve ecological variables, but there are currently only 11. There could be up to 14 in this case because I suggest including the categorical variables, but some may be removed if correlated. Just be consistent with text and table captions.
• Figure 1 title – change control to random plots.
• Figure 1 – it is difficult to distinguish between the locations at the current resolution – I count around 40 observations but know there are double that number. Adjusting the different shapes could help, or perhaps zoom in a bit more? The colors could also be difficult to distinguish between for some readers.

Experimental design

Objectives - The objectives need to be clarified and fine-tuned based on the methodology used in the study. Based on my understanding, I think that there are just two objectives: 1) identify the habitat features associated with P. mangshanensis occurrence across the study area 2) determine if the association with these features varies across season. The approach does not allow for the investigation of the actual mechanism underlying these associations. The final statement in the last paragraph is extremely broad and should be revised to include testable predictions related to these objectives.

*Note that in my rephrasing of the objectives I use the term association and occurrence, not habitat selection. We cannot simply assume that where we find an animal (occurrence) is equivalent to selection or preference, unless we have detailed behavioral data to back the observations up. I think that given the nature of the data here, stick with occurrence and association.

Methods – My major concerns with this section of the manuscript include 1) aspects of data collection and lack of sufficient detail, which caused me to pose a number of questions while reading and 2) approaches used to analyze the data. A great strength - the use of active voice for the majority of this section – there are only a few places that are passive. I recommend active voice for clarity, and the authors did a great job here. I have organized my comments and questions according to the different subsections below.

Study Area: Please include the names of the dominant tree and shrub species here.

Survey Methods: Provide a citation for the type of transect surveys conducted. Provide detailed methodology for repeatability of the transects – e.g. length, number of surveyors, initial placement, were they time-constrained, etc. Out of curiosity, were any of the individuals observed over the 4-year period recaptures, or were the 83 locations the only observations? If you had more observations, how did you determine which locations to use? I am really impressed with the sample size for such a rare snake!

My greatest confusion and issue relate to lines 92 – 94. Were the 83 locations gathered throughout 2012-2016, but the habitat data measured later in 2015 and 2016? I checked the raw data, but there are no dates for the individual snakes. If this is the case, I assume that you revisited the locations during the time of year when the individual was initially encountered, but several years later? As currently written, it seems there are different plots picked for each season and then surveys conducted within the plots themselves during those times. These details need to be laid out more clearly. After rereading several times, I think that the snake observations predate collection of the habitat data. If this is the case, there are several potential issues. While some habitat variables remain consistent over time (e.g. elevation, slope location), others likely vary (e.g. aspects associated with vegetation including canopy cover and height). If the study area is such that changes would be negligible, provide support for this. Otherwise, you need to acknowledge that such variables have likely changed since the individuals were detected.

Pick one term, either random or available for the “paired” plots and use throughout the rest of the manuscript. I think that random may be more appropriate based on the difficulty in truly describing what is available to individuals.

Habitat Variables: I am curious about your previous research – I think that it would be useful to summarize related projects for the reader. For example, which specific aspects of previous research influenced you to include certain habitat measures in this particular study? If exploratory investigations revealed tendencies for this species to occur within certain habitat types, for example, this is also useful to point out in the second to last paragraph of the introduction. The reader would then realize that while associations between the species and habitat have been observed, the details had not been rigorously investigated prior to this manuscript. Remove the numbers that precede each of the habitat variables and write out each as their own descriptive sentence. I suggest this simply because there are many of them.

Please provide clarification with regards to the following variables: gravel (how did you determine the cutoffs for the three categories? This detail could be added to table S1 or the text), slope aspect (where exactly did you walk the slope? At the precise location of the snake or across the entire plot?), slope location (what were the specific elevational cutoffs for the three categories?) distance to water (was the distance calculated from the center of the plot, and does water refer to any type, or a specific category?), fallen log density (did the logs also fall within a specific length?), canopy cover (how did you estimate vertical projection?), and deciduous cover (how did you specifically measure this? via a visual estimate?).

I don’t think that table S1 is necessary if you keep the details in the text. Either move all details to the table or add missing details to what is already in the text. If you keep the table, I think that it would be interesting to include the proportions of used and available observations that fell within each category.

Specific comments:
• Line 106 – Include the word categorical before habitat variables.
• Line 106 – I suggest just using aspect for this categorical variable as slope and aspect are two different features of the landscape. Or, you could write out aspect of the slope.

Statistical Analyses: I have several concerns with the approaches used to analyze the data, and some of the tests are redundant and can be removed. My main issue is that I don’t think that the current methodology accurately compares habitat use across seasons. Please note that I am not a statistician, and there are likely other ways to analyze the data, but I suggest an approach that I am familiar with and have used for similar data. Perhaps another reviewer or the associate editor recommend a different strategy. In addition, for the logistic regression model, the assumptions of independence are violated because the used and available plots are paired, so plot pair would need to be incorporated as a random effect.

1) To truly evaluate if habitat use varies across seasons, you need to test the effect of season in a single model. Currently, you are comparing used versus available for each season separately. Therefore, you cannot compare the relative support of the models to one another because the models are comprised of different subsets of data. In addition, when running the models separately, you reduce your power due to lower sample sizes, increasing the likelihood that effects are due to chance. The general rule of thumb is to have at least 10 samples for each variable that you include in a model. When running the models separately, there are 14 total variables, but only up to 60 observations. If you include all of the observations in a single regression analysis, then you have a sufficient sample size for the number of variables you wish to test.

2) The approach that I suggest: Run a multinomial logistic regression model with a random effect to explore if habitat associations of individuals if any and which variables are associated with the used versus the available locations (why I refer to this as objective 1 in my comments above – are the snakes selecting for certain habitat features compared to what is available to them in general), and if these associations varied across the seasons (objective 2 above). Include all the data, with 4 response variable categories, the available or random locations and then snake observations grouped by season (0 = all available or random points, 1 = spring locations, 2 = summer locations, and 3 = fall locations). Use the same approach for model evaluation as you describe to identify the best-fit models but run an ANOVA on the best-fit models to view P-values and to evaluate the significance of each explanatory variable and associated coefficients to show the direction of the relationship (Menard 2002). This will produce a table similar to table 1, but with all available data combined in one column, and will include the associated coefficients. Significant variables in the available column reveal differences between the used vs. available locations in general, and significant variables within each month group show if any seasonal differences exist. You will need to recalculate the means for the used plots.

I have never used discrimination analysis, so I cannot comment on the appropriateness of its use here, but I think that using the approach I have described above and looking at which variables are significant for the random points should get at this.

Running all of the paired tests results in multiple comparisons, and as such, some differences may be due to chance alone. This is why I suggest running the regression model instead.

3) Before deciding which variables to include in the regression model, run the Pearson correlation tests to identify and remove correlated variables. You explain doing this, just move this information to the first paragraph. You also want to include the categorical habitat variables, which the regression analysis can accommodate. Just be sure to check if any are correlated first (use Chi-square test of independence) and be aware that you need to calculate dummy variables due to more than just two categories.

Specific comments:
• How did you build your regression models? For example, did you use a forward-selection backward-elimination approach?
• Were any of the variables not normal, and if so, did you transform the data, and how?
• % correct classified included in table 2 is not mentioned in the methodology section.

Validity of the findings

Results – I think that the reason some variables detected as significant during paired comparisons are not included in the best-fit models comes down to the problem of multiple comparisons and sample size. The first time I read through the manuscript, I was confused about the differences in table 1 and table 2 in terms of variables significantly associated with snake presence.

Specific comments:
• Lines 165-169 – Change to: analysis revealed a positive association between fallen log density and snake presence and a negative association between…
• I suggest listing the variables in order of their significance OR alphabetically and then be consistent across all sections using this same approach.
• State the direction or relationship of the variables when comparing random locations with occupied ones.
• Line 193 – change proving to suggests. I agree that it is likely that the snakes are actively selecting the logs, but these data show associations. If radio-telemetry or mark-recapture data revealed snakes repeatedly on logs, then that would reveal actual selection or preference.
• Line 201 – According to the current results in table 1, shrub height is also significant in all three seasons.
• Lines 204-211 – If you decide to keep this analysis, could you provide more clarity on what these results mean? As I have previously mentioned, I am unfamiliar with the approach, and this is confusing to me.
• Figure 1 title – change control to random plots.
• Figure 1 – it is difficult to distinguish between the locations at the current resolution – I count around 40 observations but know there are double that number. Adjusting the different shapes could help, or perhaps zoom in a bit more? The colors could also be difficult to distinguish between for some readers.
• Figure 2 – Wonderful addition to include these images! What a stunning species!

Discussion – Be clear in the discussion when referring to other studies versus data reported here within all paragraphs.

1) The first paragraph needs some improvement. Again, these data do not show preference, but rather association. There seem to be two separate arguments within this paragraph that just need to be clarified. The snakes are ambush feeders – they lie in wait until prey appear – but then, they may use caudal luring to attract the prey at that point. Their whole body doesn’t resemble a food item of their prey (line 240), unless the prey they consume also consume lichens, rather their tail does. That is the body region that resembles a grub. If they use a strategy similar to Copperheads (a species that I have worked with), their camouflage allows them to blend into the habitats traversed by their prey during the preys’ foraging movements, thereby potentially increasing the likelihood of an encounter. Maybe I am being nit-picky here but am trying to provide suggestions on how to help clarify the points you are making.

Specific comments:
Line 122 – change shelter to refugia.
Lines 228-229 – you can’t state the snakes were giving logs priority unless you have detailed telemetry data or mark-recapture observations, in my opinion. You CAN say that field observations confirmed strong associations of individuals with the logs.
Line 241 – Based on other literature that I have read about this species, scientists state lichen not moss.
Line 252 – So you do have telemetry data? Very cool! Share this fact with the reader in the methods when you explain why you selected the variables that you did. This likely explains why you state “snakes preferentially crawled near logs” on line 230, so included based on unpublished telemetry data there as well (if it is unpublished, otherwise provide a reference to that study).
Lines 248-255 – Very interesting that you did not observe the snakes drinking water, but visually noticed the increased presence of small mammals!

Conclusions – 2 out of the 3 recommendations are great! I don’t clearly see the connection between the first recommendation and the results reported here. Is the increased likelihood of human/snake contact due to the fact that the snakes are using more “open” areas in the spring?

Additional comments

Figure 2 – Wonderful addition to include these images! What a stunning species!

---

## Round 0.2 · Minor Revisions

Thank you for your thorough revision and response to the reviewers. This is a much improved manuscript that more clearly articulates your study and its implications.

The reviewers have identified a small number of remaining issues that should be addressed before the manuscript is accepted. For the most part, these issues include additional details for the reader and some edits to various word choices. Note that Reviewer 2 has attached an annotated document for you.

I look forward to the next version of your manuscript.

·

Basic reporting

Some details could be refined to more clearly communicate meaning.

Experimental design

I am not qualified to evaluate the details of the anlaysis.

Validity of the findings

Ten meters is a substantial distance and includes a lot of hiding spots. A snake under dense cover is going to be more detectable than a snake in the open next to or on top of a log. Given the focus of the manuscript on identifying important microhabitat features, it is important to consider how this variation in detection probability influenced observations and why I brought it up in my initial review. I’m glad to see there is now some general information in the discussion about how detectability may have affected observations, but it is worth considering including some specifics based on the results. For example, is a snake by a log more detectable? How would that have affected data. The authors now state that a number of measures were incorporated to address detection issues (for example, standardizing start times and observation method) but I’m not sure how they are relevant. Ideally detection probability is incorporated into analysis, but if this is not possible, then it needs to be discussed to a greater extent.

Further justification is needed as to why it is acceptable to quantify microhabitat at a site up to three years after a snake was spotted there. Stating that it reveals their preferences only to “a certain extent” is not informative.

There is still no information about how/why it is appropriate to quantify habitat selection at only one spatial scale given most snakes likely select habitat at multiple spatial scales. Is it possible that something important or relevant to management is overlooked, why or why not?

Specifically how many individual snakes were observed?

Promoting the value of tourism has little to do with the study so I’m not sure why it is a part of a conclusion.

Additional comments

The manuscript is much improved.

·

Basic reporting

I find no problems with the basic reporting. The article is well written in excellent English. I have provided some additional suggestions on word choice below. The literature cited and background/context is sufficiently developed in the Introduction. The tables, figures, and raw data are appropriate. The results and hypotheses are connected and relevant.

Experimental design

The methods are improved from the previous submission and adequately address the concerns and suggestions of reviewers. I have not used random forest methods, so I don’t feel qualified to comment on the specifics of this technique. The supplemental tables allow for replication in the field if needed. Please see attached document for specific comments.

Validity of the findings

The conclusions are well stated, are linked to the original research question, and are limited to supporting results. I found no indication of social spiders in the raw data. I feel the authors should include additional discussion on lag time (in years) between early snake detections and habitat surveys, as well as a more detailed discussion of the potential impact of imperfect detection on their surveys. Overall, I suggest changing the use of "occurrences" to "detections" throughout. Please see attached document for specific comments.

Additional comments

I enjoyed reading the revised manuscript.

---

## Round 0.3 · accepted · Accept

Thank you for your thorough and thoughtful revision - I am excited for the publication of your manuscript.

There are only a few minor tense issues to deal with. For example, in the sentence "Our approach of using transect survey to discover Mangshan pit viper provides" from the first paragraph of your discussion should have the word "surveys" rather than "survey". And in the second sentence of that paragraph, "detections was" should be changed to "detection was". In the final steps prior to publishing, please make these changes and carefully check for any other tense issues throughout.